# The pharynx of the stem-chondrichthyan *Ptomacanthus* and the early evolution of the gnathostome gill skeleton

Richard P. Dearden [1], Christopher Stockey[1,2] & Martin D. Brazeau[1,3]

The gill apparatus of gnathostomes (jawed vertebrates) is fundamental to feeding and ventilation and a focal point of classic hypotheses on the origin of jaws and paired appendages. The gill skeletons of chondrichthyans (sharks, batoids, chimaeras) have often been assumed to reflect ancestral states. However, only a handful of early chondrichthyan gill skeletons are known and palaeontological work is increasingly challenging other presupposed shark-like aspects of ancestral gnathostomes. Here we use computed tomography scanning to image the three-dimensionally preserved branchial apparatus in *Ptomacanthus*, a 415 million year old stem-chondrichthyan. *Ptomacanthus* had an osteichthyan-like compact pharynx with a bony operculum helping constrain the origin of an elongate elasmobranch-like pharynx to the chondrichthyan stem-group, rather than it representing an ancestral condition of the crown-group. A mixture of chondrichthyan-like and plesiomorphic pharyngeal patterning in *Ptomacanthus* challenges the idea that the ancestral gnathostome pharynx conformed to a morphologically complete ancestral type.

[1] Department of Life Sciences, Imperial College London, Silwood Park Campus, Buckhurst Road, Ascot SL5 7PY, UK. [2] Centre for Palaeobiology Research, School of Geography, Geology and the Environment, University of Leicester, University Road, Leicester LE1 7RH, UK. [3] Department of Earth Sciences, Natural History Museum, London SW7 5BD, UK. Correspondence and requests for materials should be addressed to M.D.B. (email: m.brazeau@imperial.ac.uk)

The pharynx of jawed vertebrates (gnathostomes) plays a fundamental role in feeding and respiration. Furthermore, the skeletal pharyngeal arches are central to theories on the origin of jaws and shoulder girdle[1–4], making the pharynx of interest to those seeking the origins of these structures. Information with which to establish its ancestral form can be found in the anatomy of fossilised Palaeozoic gnathostomes: members of now extinct gnathostome groups, as well as early members of the two modern gnathostome clades (the cartilaginous and bony fishes). However, fossils that actually preserve the fragile pharyngeal skeleton are vanishingly rare. Major differences, such as that between the compact, subcranial pharynx of osteichthyans (bony fishes) and the caudally extended pharynx of most chondrichthyans (cartilaginous fishes), remain difficult to reconstruct. Similarly, the polarity of a wealth of phylogenetically important differences in the arrangement of pharyngeal elements are poorly understood. Palaeozoic taxa in which we do understand pharyngeal morphology can prove decisive in resolving these problems and allow us to arbitrate on scenarios of the anatomical and functional evolution of the pharynx[5–7].

Views of pharyngeal evolution have long been coloured by a perception that modern chondrichthyans are especially primitive[2]. Palaeontological discoveries have gradually overturned this view, revealing that the superficially basic chondrichthyan anatomy of cartilages and simple scales is in fact derived states[8–11]. However, chondrichthyans have continued to be used as proxies for plesiomorphic gnathostome conditions in the pharynx[2,3,12], as well as fins[13], jaws[2,14] and genomes[15]. Recent investigations using computed tomography scanning have revealed a spate of new data on the gill skeletons of early gnathostomes[5,6,16]. However, chondrichthyan examples display a pharyngeal skeleton already showing an elasmobranch-like caudal extension. Combined with phylogenetic uncertainty amongst early chondrichthyans this leaves many questions about primitive early chondrichthyan pharyngeal structure open.

Here we use computed tomography to reveal the pharyngeal morphology in an acanthodian-grade stem-chondrichthyan: *Ptomacanthus anglicus*. *Ptomacanthus* is from the Lochkovian (419.2–410.8 Mya) of Herefordshire, UK[17–19], making this the earliest three-dimensionally articulated branchial skeleton known in a gnathostome. The pharynx of *Ptomacanthus* is organised like that of osteichthyans, lying entirely beneath the neurocranium and covered by a bony operculum. This contrasts with the caudally extended pharynx of most chondrichthyans. Meanwhile the patterning of the skeletal elements in the pharynx of *Ptomacanthus* comprises a combination of plesiomorphic and chondrichthyan-like character states. We incorporate this new information into a phylogenetic analysis, allowing us to constrain the elongation of the chondrichthyan pharynx to a specific region of the stem-group. We reconstruct an ancestral gnathostome pharynx which displays a mixture of chondrichthyan-like and plesiomorphic states. *Ptomacanthus* supplies a key reference with which to interpret other early chondrichthyan branchial skeletons, to decipher comparative pharyngeal structure across modern gnathostome groups, and to address current phylogenetic uncertainty deep in the chondrichthyan stem-group.

## Results

**Pharyngeal morphology in *Ptomacanthus*.** Almost the entire visceral skeleton of *Ptomacanthus* is preserved: Meckel's cartilages, the hyoid arch, and five branchial arches on either side (Figs. 1, 2, Supplementary Figs. 1, 3). There is a single, median basihyal, followed by a paired series of ceratohyals and ceratobranchials. Dorsally, on the anatomical left side, there is a complete series of epibranchials preserved, and three partially preserved epibranchials on the right side (likely corresponding to arches 1, 2, and 4). On the left side, a series of largely indistinct mineralisations are present and correspond to pharyngobranchials. There is an empty field between the ceratobranchials, suggesting basibranchial or hypobranchial cartilages were present but not mineralised. However, the ceratobranchial of the fifth arch on the anatomical left-hand side terminates in a particularly large crescentic structure, possibly corresponding to a hypobranchial or basibranchial. The front-most arches, including the first, are extremely crushed and difficult to distinguish from one another, particularly distally. Posteriorly the three-dimensional nature of the arch elements can be seen more clearly (Fig. 2, Supplementary Fig. 1d).

Between the ceratohyals of *Ptomacanthus* lies a trapezoidal, flattened basihyal (Fig. 2, Supplementary Fig. 1a). It appears to consist of globular calcified cartilage (Supplementary Fig. 2). The anterior corners of the basihyal are closely associated with the ceratohyals, overlying the tip of the left ceratohyal and lying laterally to the right. This suggests that it lay dorsal to the ceratohyals in life as in chondrichthyans[5]. Posteriorly this basihyal lies next to, and presumably also articulated with, the anteriormost branchial arch (Fig. 2, Supplementary Fig. 1a). There is no evidence for unpaired mineralisations posterior to the basihyal. The close proximity of the first ceratobranchials and the basihyal suggests that hypobranchials were absent in the first branchial arch, as seen in *Acanthodes* (see Discussion) and *Gladbachus*[6].

The ventral halves of all five of the branchial arches in *Ptomacanthus* are separate, and comprise a parallel series of long, curved, posteriorly grooved ceratobranchials (Fig. 2b, c, Supplementary Fig. 1a). The arches taper to meet the upper branchial elements – this proximal region is more intact on the anatomical right-hand side of the specimen where a canal running oblique to the direction of the arch is visible on the first and second arches (Supplementary Fig. 1c). On the anatomical right hand side of the branchial skeleton each lower element is divided by a break oblique to the element's direction (Fig. 2b, c, Supplementary Fig. 1a). On the anatomical left-hand side this division is not apparent, but does correspond with a point at which the arches sharply kink. Distally the arches terminate with box like ends.

Each epibranchial is posteriorly grooved (Fig. 2b, Supplementary Fig. 1a) and curves ventrally to meet its corresponding ceratobranchial. The epibranchials lack any posterior flanges as in some chondrichthyans[6]. The distal ends of the elements are expanded and appear to link antero-posteriorly – we interpret these as pharyngobranchials, although any separation from their corresponding epibranchials is unclear. The pharyngobranchials are antero-posteriorly ridged and the more anterior elements are smaller, with the anteriormost element being small and capsular in shape. The fifth epibranchial lacks an expanded upper part, and instead articulates with the posterior end of the fourth pharyngobranchial (Fig. 2, Supplementary Fig. 1d, g). On the anatomical right-hand side of the specimen the first, second and third elements are largely preserved as moulds on the surface of the matrix (Supplementary Fig. 3). However, on this side some remnants are left of several epibranchials, and both a posteriorly displaced fourth epibranchial, and a small anterior pharyngobranchial probably corresponding to the left pharyngobranchial 1 are present within the matrix (Fig. 2a, Supplementary Fig. 1e, f).

The dorsal elements of the branchial skeleton are seen mainly on the anatomical left-hand side. The articulations of these dorsal elements with the braincase cannot be directly examined. However, the relatively undisrupted articulation of the elements shows that the dorsal ends of the three anterior-most arches are aligned along the margins of the parachordal plates. This means that they likely had their articulations on or next to the braincase (Fig. 1, Supplementary Fig. 3).

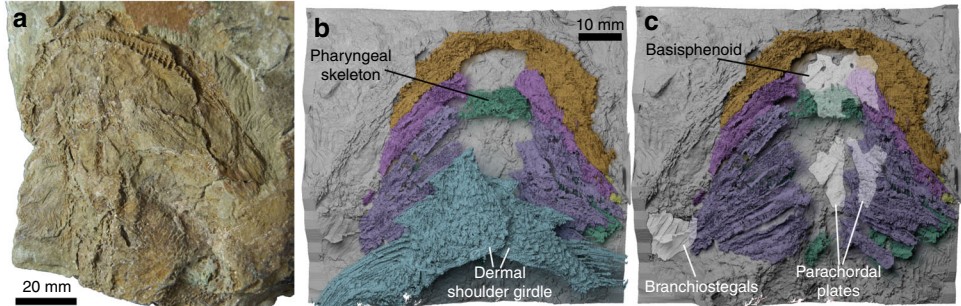

**Fig. 1** *Ptomacanthus anglicus*, specimen NHM P.24919a. **a** Photograph. **b** Virtual rendering of the pharyngeal skeleton and dermal shoulder girdle in ventral view, with the moulded surface of the matrix in the background. **c** Virtual rendering of the pharyngeal skeleton in ventral view, with the moulded surface of the matrix in the background, and the positions of the braincase (as interpreted by Brazeau[18]) and branchiostegals overlain

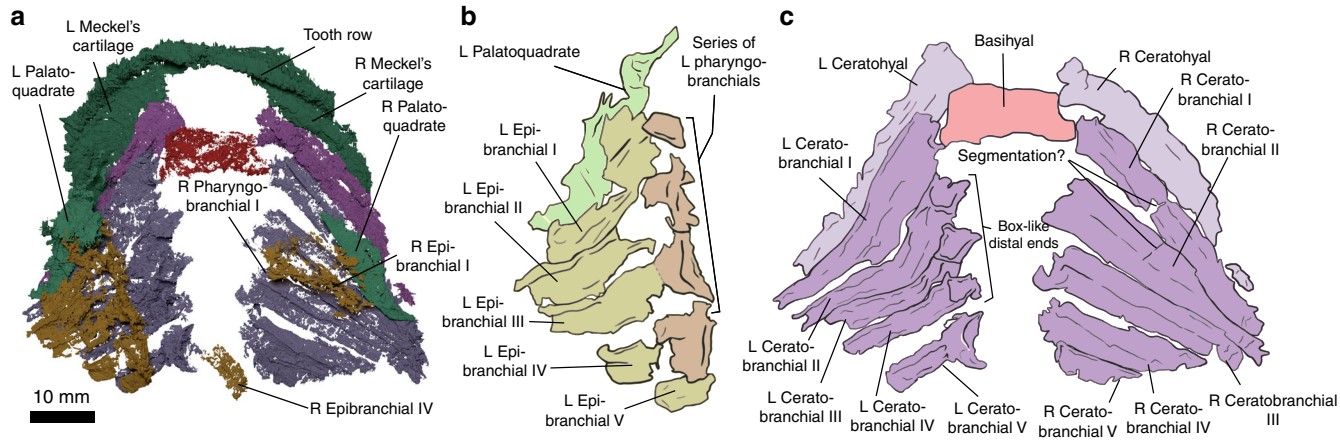

**Fig. 2** The pharyngeal skeleton of *Ptomacanthus anglicus*. **a** Virtual rendering in dorsal view. **b** Drawing of dorsal branchial skeleton from the anatomical left hand side. **c** Drawing of ventral branchial skeleton. Colour scheme: red, basihyal/branchial; purple, ceratohyal (light shade)/branchial (dark shade); yellow, epibranchial; orange, pharyngohyal/branchial; green, mandibular arch

**Phylogenetic analysis**. Based on these observations, we updated the phylogenetic matrix of Coates et al.[6]. In the strict consensus tree there is almost no resolution of the chondrichthyan stem-group (Supplementary Fig. 4a). However, in the Adams consensus some consistent branching patterns are revealed. Importantly, the stem-chondrichthyan *Gladbachus* has moved crownwards on the chondrichthyan stem-group relative to *Ptomacanthus* (Supplementary Fig. 4b). Filtering these trees reveals that all trees conform to the topology (Stem-gnathostomes(*Ptomacanthus*(*Gladbachus*(Crown-chondrichthyans))). The same topology is recovered by the Bayesian analysis (Supplementary Fig. 5). This contrasts with Coates et al.[6] who recovered *Gladbachus* intercalating the acanthodian grade, as the sister group to all total-group chondrichthyans except the Acanthodii. Our placement of *Gladbachus* is supported by comparative gill arch characters that have been modified or introduced here (see Supplementary Note 1). All acanthodians are recovered as stem-chondrichthyans, with *Ptomacanthus* and other climatiid taxa paraphyletic relative to diplacanthids, ischnacanthids and acanthodids, including *Acanthodes* (Supplementary Figs. 4 and 5). Also recovered in a relatively crownwards position on the stem-group are the seemingly more shark-like taxa *Lupopsyrus*, *Obtusacanthus*, *Kathemacanthus* and *Pucapampella*, and *Doliodus* is recovered as the sister taxon to crown-chondrichthyans. Within the crown-group relationships symmoriiformes are recovered as stem-holocephalans, and hybodonts, ctenacanths and xenacanths as stem-elasmobranchs (Supplementary Figs. 4 and 5).

## Discussion

The compact pharynx of *Ptomacanthus* allows us to constrain the evolution of a chondrichthyan-like pharynx to a specific region of the chondrichthyan stem-group. This fits into an emerging picture of early gnathostome relationships, with mounting evidence placing the entire acanthodian grade in the chondrichthyan stem-group[6,9,20]. Given the presence of a ventral, compact pharynx in both osteichthyans (eg. *Raynerius*[16]) and stem-gnathostomes (eg. *Paraplesiobatis*[21]) this means that an osteichthyan-like pharynx can be placed at the gnathostome crown-node with a reasonable degree of confidence. The ostensibly osteichthyan-like condition in extant holocephalans is likely homoplasious, given the presence of an elasmobranch-like pharynx in the stem-chondrichthyan *Gladbachus*, and the probable stem-holocephalan *Ozarcus*[5,6].

Coates et al.[6] recovered the Devonian chondrichthyan *Gladbachus*, which clearly shows an extended elasmobranch-like pharynx, amidst the acanthodian grade on the chondrichthyan stem-group. Contrastingly our reworking of their dataset and the addition of new information on *Ptomacanthus* to it results in *Gladbachus* lying crownwards of the acanthodian grade, in part as a result of the morphology of its pharynx. This position is consistent with a host of other characters grouping *Gladbachus* with the chondrichthyan crown-group to the exclusion of acanthodian-grade taxa, including extensively calcified cartilage, the absence of any kind of large dermal cranial ossification, and the absence of paired fin spines[6]. This could restrict the evolution of an elasmobranch-like pharynx to somewhere between *Ptomacanthus* and *Gladbachus* in the chondrichthyan stem-group. However, the pharynx of the acanthodian-grade stem-

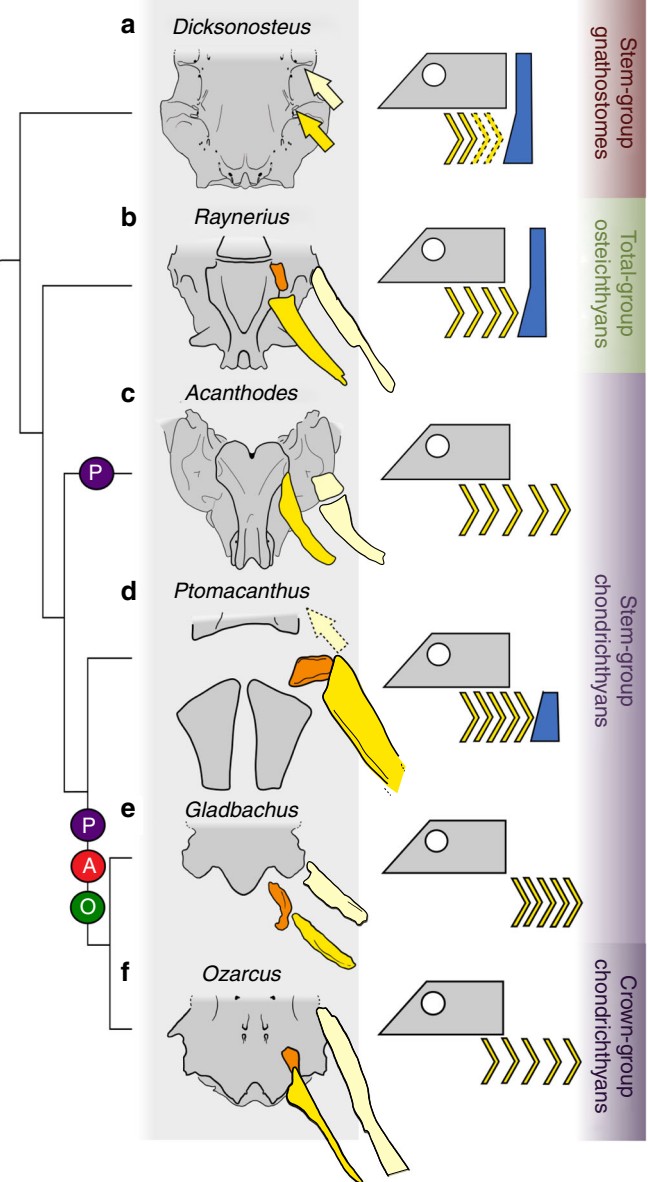

**Fig. 3** Postero-anterior extent of branchial skeleton in early gnathostomes. The phylogenetic tree is congruent with the topology described in the results (Supplementary Figs. 4, 5). Changes in position are reconstructed: P indicates a shift of the posterior extent of the branchial skeleton, A indicates a shift of the anterior extent, O indicates the appearance or loss of a bony operculum. Colour scheme: grey, neurocranium; pale yellow, hyomandibular; bright yellow, epibranchial; orange, pharyngobranchial; blue, dermal shoulder girdle. Reconstructions redrawn from refs. 5,6,20,22,25,46. The redrawn illustration from figure 8 of Maisey[46] is used with permission from the American Museum of Natural History

chondrichthyan *Acanthodes* and relatives (acanthodiforms) appears to show the extended elasmobranch-like condition[6]. Our recovery of *Acanthodes* in a grade subtending other stem-chondrichthyans including *Ptomacanthus* means that it is equally parsimonious to assume that the condition either in *Acanthodes* or *Ptomacanthus* is apomorphic.

This impasse can be resolved by closer examination of the two pharyngeal conditions in jawed vertebrates, which reveals that they can be split into three morphological variables (Fig. 3). The first of these is the anterior extent of the branchial arches: in osteichthyans (eg. *Raynerius* Fig. 3b; ref. [16]) and stem-gnathostomes (eg. *Dicksonosteus* Fig. 3a; ref. [22,23].) this lies anteriorly to the parachordal plates, where the anteriormost arches articulate on the basicranium. In the elasmobranch condition there is no articulation between the neurocranium and the anterior branchial arches, and the anteriormost branchial arch lies posterior to the otic capsule (eg. *Gladbachus* Fig. 3f; ref. [6]). *Ptomacanthus* exhibits a condition more similar to osteichthyans, with a relatively anterior extent of the pharynx (Fig. 3d). *Acanthodes* also appears to exhibit this condition (Fig. 3c), with Miles[7,24] interpreting facets in the basioccipital ossification as having articulated with anterior branchial arches. This aspect of the elasmobranch condition can be constrained on the chondrichthyan stem-group between *Ptomacanthus* and *Gladbachus*.

The second aspect that makes a pharynx elasmobranch-like is the degree of posterior extension. In osteichthyans (eg. *Raynerius*[16]) and stem-gnathostomes (eg. *Paraplesiobatis*[21]) the arches lie mostly ventrally relative to the neurocranium. Contrastingly in the elasmobranch condition they extend well caudally (eg. *Ozarcus*[5]). *Ptomacanthus* corresponds with the osteichthyan condition in this respect, with the branchial arches not extending far past the parachordal plates (Figs. 1 and 3). In *Acanthodes*, however, the pharynx extends far posterior to the neurocranium, similarly to elasmobranchs[25]. The conditions in *Acanthodes* and *Ptomacanthus* could be equally parsimoniously resolved as being homoplasious, and in most other acanthodian-grade taxa the condition is unclear. A seemingly elongated dermal gill-covering in taxa such as *Euthacanthus*[26,27] and *Mesacanthus*[26] is perhaps indicative of an *Acanthodes*-like extended pharynx, but the flattened nature of the specimens makes judging the relative positions of parts of the head difficult. One specimen of *Diplacanthus* clearly shows the branchial skeleton lying ventrally relative to the braincase[26]. If the extended condition in *Acanthodes* is indeed apomorphic it is possible that, as Miles[28] suggested, it could be linked to its anguilliform body shape: like *Acanthodes*, eels have an elongate branchial skeleton and a posteriorly displaced shoulder girdle[29].

The final defining aspect of an elasmobranch-like pharynx is the absence of an operculum. In osteichthyans and placoderms bony plates cover the gill arches as part of a functionally linked complex of dermal bones enclosing the head and shoulder girdle (eg. *Mimipiscis*[30], *Coccosteus*[31]). Contrastingly in living elasmobranchs a series of gill slits are present with no macromeric operculum. Similar gill slits are also often assumed to have been present in shark-like fossil chondrichthyans with an otherwise elasmobranch-like pharynx (eg. *Ozarcus*[5]). However, a fleshy operculum like that in living holocephalans may have been present in some early chondrichthyans[6,32–34]. A bony operculum is present in *Ptomacanthus*[17,19] as well as in *Acanthodes*[26] (although much reduced). Bony operculae are also present in many other acanthodian-grade stem-chondrichthyans, although often only forming a partial cover, with dermal ornament demarcating parts of gill slits posteriorly and dorsally (for example, in *Climatius* and *Euthacanthus*[26]). Other stem-chondrichthyans, however, such as *Brochoadmones*, *Kathemacanthus*, *Doliodus* and *Gladbachus* clearly lack a bony operculum[6,35–38]. Bony operculae appear to have been lost in the chondrichthyan stem-group somewhere between *Ptomacanthus* and these taxa. However, retention of a fleshy operculum cannot be entirely ruled out.

Mapped onto our phylogenetic tree, these character states show the piecemeal evolution of an elasmobranch-like pharynx through the chondrichthyan stem-group. *Ptomacanthus* demonstrates the osteichthyan-like condition in all three aspects of pharyngeal morphology: these are then lost on the stem-group prior to the divergence of *Gladbachus* and crown-group chondrichthyans.

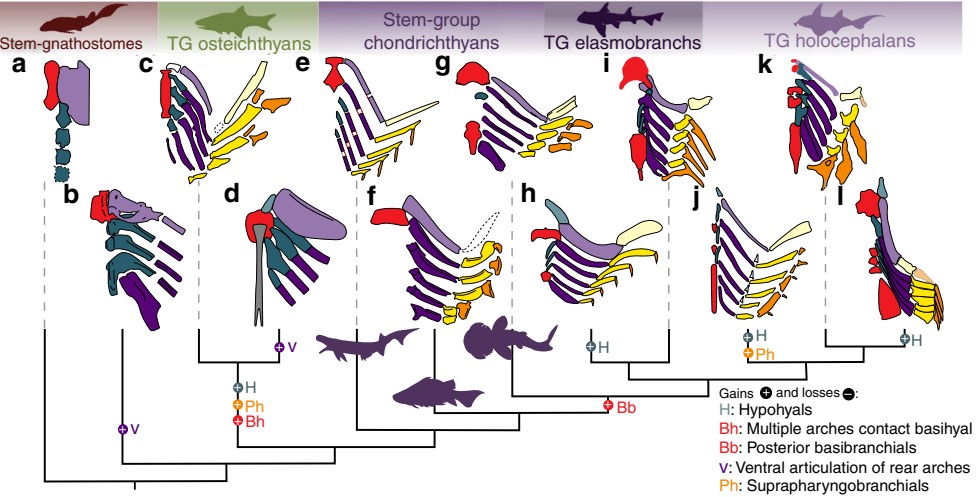

**Fig. 4** The phylogenetic distribution of branchial arch morphologies in gnathostomes. The scheme of relationships is based on the strict consensus tree derived from the parsimony analysis described in the methods. Character changes of selected pharyngeal arch characters from our analysis are plotted using an ACCTRAN optimisation. Taxa shown are **a** *Cowralepis*[47], **b** *Paraplesiobatis*[21], **c** *Raynerius*[16], **d** *Glyptolepis*[48], **e** *Acanthodes* (data from refs. [17,30]), **f** *Ptomacanthus*, **g** *Gladbachus*[6], **h** *Triodus*[49], **i** *Scyliorhinus*[5], **j** *Ozarcus*[5], **k** *Callorhinchus*[5], **l** *Debeerius*[14]. Colour scheme: as in Fig. 2, with addition of: light turquoise, hypohyal; dark turquoise, hypobranchial; grey, urohyal; pale orange, pharyngohyal. Silhouettes from refs. [19,30,50] and Phylopic. Phylopic images of *Bothriolepis* and *Callorhinchus milii* were submitted by Ghedoghedo and Tony Ayling (vectorised by Milton Tan) respectively. Both were distributed under a Creative Commons Attribution-ShareAlike 3.0 Unported license (https://creativecommons.org/licenses/by-sa/3.0/). The redrawn illustration from figure 7 of Heidtke et al.[49] is used with permission from the Landesamt für Geologie und Bergbau Rheinland-Pfalz

The shape and arrangement of components of the pharyngeal skeleton varies in jawed vertebrates, providing morphological characters that can be diagnostic for their relationships[5,39]. *Acanthodes*, because of its rare endoskeletal preservation, has been key to interpreting these characters in the very earliest gnathostomes. However, multiple interpretations of the branchial skeleton in *Acanthodes* exist on account of its unusual pattern of segmentation of largely indistinct tube-like bones[7,26,30,40,41]. *Ptomacanthus* helps evaluate the competing interpretations of *Acanthodes*, adding clarity to the ancestral complement of elements in the gnathostome gill skeleton.

When contextualised *Ptomacanthus* displays a combination of derived and plesiomorphic pharyngeal traits (Fig. 4). The presence of a basihyal contacting the hyoid and first branchial arch, and the absence of hypohyals and a mineralised posterior basibranchial adds credence to the reconstruction of *Acanthodes* by Gardiner, which displays identical conditions[30]. These states occur in both the chondrichthyan stem-group and in the gnathostome stem-group[21] suggesting that these morphologies of the ventral pharynx were likely present at the gnathostome crown-node (Fig. 4). The articulation of the ventro-posterior branchial arches seen in *Paraplesiobatis* and sarcopterygians (Fig. 4) is clearly absent in *Ptomacanthus*, consistent with either homoplasy or its plesiomorphy for gnathostomes. The dorsal branchial skeleton of *Ptomacanthus* shows marked similarities to those of other chondrichthyans[5,6], with an antero-posterior chain of dorsally ridged pharyngobranchials and five epibranchials, the posteriormost two of which articulate on the same pharyngobranchial. This last condition is comparable to the articulation between the epibranchial IV and the fused epibranchial V and pharyngobranchial IV in Recent chondrichthyans[12]. This implies a plesiomorphic crown-gnathostome pharynx which, rather than being exclusively chondrichthyan-like, shows a mélange of osteichthyan-like and chondrichthyan-like morphologies.

It has been assumed, implicitly or otherwise, that the ancestral pharyngeal skeleton must have been morphologically complete, with one of every part that we observe in branchial arches (hypo-, cerato-, epi-, pharyngo-, etc.) in every other pharyngeal arch. This has formed the basis of attempts to reconstruct the evolution of the gnathostome jaw suspension using a supposedly morphologically complete hyoid arch with pharyngo- and hypohyals[14], as well as in the sporadic revival of the aphetohyoidean hypothesis – the expectation that we should observe a fully functional hyoid gill slit in the earliest gnathostomes[5,26,42]. However, neither the ancestral gnathostome branchial skeleton we reconstruct here, nor the earliest fossil gnathostome branchial skeletons show any evidence for being morphologically complete. They lack features that might be expected in such a skeleton, for example, hypohyals and pharyngohyals. They also display anatomical conditions that are difficult to interpret in terms of living gnathostome branchial skeletons. This is still consistent with a serial homology between arches. However, the earliest branchial skeletons should not necessarily be expected to be morphologically complete, and nor should apparently morphologically complete branchial skeletons necessarily be expected to represent an instantiation of a plesiomorphic state.

This new information from *Ptomacanthus* provides an important cipher for interpreting the branchial skeletons of other early gnathostomes. The reorganisation of the chondrichthyan pharynx into an elasmobranch-like configuration can now be constrained to a specific part of the chondrichthyan stem-group, between the divergences of *Ptomacanthus* and *Gladbachus* from the crown-chondrichthyan clade. *Ptomacanthus* allows us to reconstruct an ancestral branchial skeleton which lacked mineralised hypohyals, and which possessed a large median basihyal articulating with the hyoid arch and one branchial arch. Importantly, this reconstructed ancestral skeleton challenges notions of a "morphologically complete" ancestral gnathostome pharynx. While the pharynx of *Ptomacanthus* is chondrichthyan-like in some respects, many aspects of the elasmobranch pharynx are certainly derived, and care should be taken using the chondrichthyan condition as an ancestral proxy when building scenarios of pharyngeal evolution.

## Methods

**Specimen**. The specimen that we used in this study is *Ptomacanthus anglicus*[17] Natural History Museum, London (NHM) specimen P.24919a, comprising the

anatomically ventral half of a siltstone concretion containing the head and pharynx of a fish, and previously described by Miles[17] and Brazeau[18,19]. The part has been etched using hydrochloric acid, leaving moulds of parts of the endoskeleton on the surface[17]. An x-ray of the specimen revealed that the concretion held additional skeletal material within, leading to this study. The counterpart, NHM specimen P.24919b, was found to contain no significant endoskeletal material.

**Geological context**. P.24919a was originally collected from a siltstone lenticle in Wayne Herbert Quarry, Herefordshire, UK[17,43]. This locality is interpreted as having been fluvial on the basis of isotopic and sedimentological evidence[44,45]. In addition to further specimens of *P. anglicus* Wayne Herbert has yielded a number of other articulated acanthodians, as well as articulated osteostracans and heterostracans, and pieces of articulated thelodont[17,43]. The site lies within the *Rhinopteraspis crouchi* zone, and so is Lochkovian in age (419.2–410.8Mya)[43].

**Micro computed-tomography scanning**. P.24919a was scanned using a Nikon XT H 225 ST at 224 kV and 240 µA, with a 39 µm voxel size. Data were segmented with Mimics v.18 (biomedical.materialise.com/mimics; Materialise, Leuven, Belgium). Segmented models were then imported into Blender (blender.org) to acquire images.

**Phylogenetic analysis**. Both parsimony and Bayesian analyses were run, on a matrix of 88 taxa and 267 characters. This matrix was adapted from Coates et al.[6] with the addition of information on the branchial skeleton of *Ptomacanthus*, a reworking of characters related to the branchial skeleton, and the addition of two taxa - *Paraplesiobatis* and *Euthacanthus*. Details of these modifications are available in Supplementary Note 2.

The parsimony analysis was run in TNT 1.5 (Goloboff and Catalano 2016) using a parsimony ratchet with 10,000 iterations, holding 100 trees each iteration, and using TBR branch swapping. Galeaspida was set as an outgroup and the constraint (Galeaspida (Osteostraci (Mandibulate gnathostomes))) was enforced.

The Bayesian analysis was run in MrBayes v. 3.2.6. Mandibulate gnathostomes were constrained as monophyletic and Galeaspida were set as the outgroup. A flat (uniform) prior was used along with the M$kv$ model and a gamma-distributed rate parameter. We then carried out the search for 10,000,000 generations, sampling a tree every 1000 generations, and calculated a majority rule consensus tree with a relative burn-in of 25%.

**Reporting summary**. Further information on research design is available in the Nature Research Reporting Summary linked to this article.

## Data availability
The CT data that this study is based on, as well as 3D models of the described material, are available in figshare, and available here: https://figshare.com/projects/Supplementary_Information_Ptomacanthus_pharynx/59846. All other files underlying this study, including the phylogenetic dataset, additional notes, and videos of the models, are available in the supplementary information files.

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

## Acknowledgements

We thank Emma Bernard (Natural History Museum, London, UK) for the loan of the specimen and Tom Davies (University of Bristol, UK) for help and advice scanning it. Zerina Johanson and Patrick Campbell are thanked for access to and training on radiography at NHM. This work was funded by a European Research Council grant awarded to M.D.B. under the European Union's Seventh Framework Programme (FP/2007–2013)/ERC Grant Agreement number 311092. TNT was made freely available with the sponsorship of the Willi Hennig Society.

## Author contributions

R.P.D. and M.D.B. conceived the project. C.S. X-rayed specimens at the NHM, leading to the study. R.P.D. and M.D.B. collected tomographic data. R.P.D. processed the tomographic data. R.P.D. assembled and analysed the phylogenetic dataset. R.P.D. and M.B.D. interpreted the data and wrote the first draft of the manuscript. All authors discussed and commented on the final version of the manuscript.

## Additional information

**Competing interests:** The authors declare no competing interests.

