## [Peer Review File · Nature Communications]

Reviewers' Comments:

Reviewer #1:

Remarks to the Author:

This manuscript provides the first description of the visceral (branchial) skeleton of the 'acanthodian'-grade stem chondrichthyan *Ptomacanthus* from the Lower Devonian of UK. These data on 'acanthodians' were eagerly awaited in the framework of current debates about the origin of chondrichthyans and the plesiomorphic condition of the branchial skeleton in jawed vertebrates. The paper is excellently executed, and the segmentations of the CTscan images is a real technical feat, especially considering the low degree of mineralization of the perichondral lining of the elements of the branchial skeleton in these taxa. The authors provide a very interesting interpretation of the timing of the process of posterior extension of the branchial skeleton at the node "between" *Ptomacanthus* and *Gladbachus*. The conclusion is that the plesiomorphic pattern for the crown gnathostome pharynx may have been a mix of osteichthyan-like and chondrichthyan-like patterns. I would agree entirely.

Minor corrections:

p.8, l. 254: "While the condition in could be equally parsimonious..." in what? A word seems to be missing.

Caption to figure 3: "anterior extent, O indicates the appearance..." I don't see this "O" in the figure

Caption to fig. 4: "(h) Debeerius"... isn't it rather (l), as (h) already appears in the line above for *Triodus*?

Reviewer #2:

Remarks to the Author:

The gill skeleton is a key vertebrate anatomical system which provides a wealth of phylogenetic data to understand the evolution of early gnathostomes, as well as in debates on the origin of jaws or paired appendages. However, due to weak mineralization the fossil data on the gill skeleton of early gnathostomes are comparatively few, limiting our understanding of their character polarity or leaving considerable uncertainty on the ancestral complement of elements in the gnathostome gill skeleton. The description of the gill skeleton of *Ptomacanthus*, the earliest three-dimensionally articulated branchial skeleton known in a gnathostome, will help to constrain some major changes in pharyngeal architecture close to the split between osteichthyans and chondrichthyans or close to the chondrichthyan crown node. The general morphology of gill skeleton in *Ptomacanthus* is well described and very informative. As such, I recommend it to be accepted with minor revisions.

Meanwhile, I wonder that the authors should be careful of the discussions on the 'archetype', which is out-dated, and more like a jackstraw. With the updated phylogenetic scenario of early jawed vertebrates, it is normal that the previous understanding on the ancestral character combination is under change, along with the fast-emerging evidences, either from the new fossils or by means of new technique as in this study.

Regarding data accessibility, I recommend the authors to upload the stacks and all scan parameters to a public platform for download, if possible. This will help readers examine the digital restorations, which form the cornerstone of this manuscript.

l.31-32: "However little is known about the gill skeletons of early chondrichthyans....."

Note that the gill skeleton is well described in one of acanthodians (acanthodian-grade stem chondrichthyans), *Acanthodes*, and some early conventionally-defined chondrichthyans, such as *Gladbachus* and *Ozarcus*.

I.38: "a combination of chondrichthyan-like and plesiomorphic conditions" as "a combination of chondrichthyan-like and plesiomorphic gnathostome conditions".

I.116: "the ceratohyal of the fifth arch" Do you mean the ceratobranchial of the fifth arch?

I.183: Gladbachus in italics.

I.254: "While the condition in could be"; in what?

I.246-259: The arguments in this paragraph are important but not so solid. As acknowledged by the authors, the posterior extension in Acanthodes and Gladbachus + crown chondrichthyans can be equally resolved as one innovation at the node TG chondrichthyans and one reversal in Ptomacanthus. The argument "a seemly compact condition in the acanthodian Diplacanthus" is weak, considering many other acanthodians with the same phylogenetic grade as Acanthodes have the similar elongated opercular area (as indicated by the long branchiostegals, such as in Euthacanthus and Mesacanthus) to Acanthodes. The resolution of this uncertainty of character polarity needs additional fossil data close the crown-gnathostome node. This is beyond the scope of this study.

I.260-278: The presence or absence of a bony operculum is one of main topics in this manuscript. It is very helpful to readers to add some photos or illustrations of the bony operculum in Ptomacanthus and Acanthodes, at least in supplementary information.

Figure 3. No 'O' in the illustration.

Figure 4: "as in figure 6"? Do you mean "as in Figure 2"?

I.311-314: You need to rephrase this sentence.

In Figures S1-S3, the scale bars need to be added, as in Figures 1 and 2.

Min Zhu

Reviewer #3:

Remarks to the Author:

GENERAL COMMENT

Dear Authors,

I find this MS interesting and the studies well performed.

The manuscript details a study of the gill skeletons preserved in a fossil (415 m.y. old) STEM chondrichthyan (Ptomacanthus). This specimen was studied in the past by several authors and demonstrated as a key fossil to understand polarity of characters and interrelationships of mayor groups of early gnathostomates. In the present mns. Dearden & Brazeau use computed tomography scanning to study new anatomical structures in the specimen (i.e. the branchial apparatus). The new data contribute to change the erroneous view of extant chondrichthyans as indirect means for plesiomorphic gnathostomate pharynx conditions. Contrarily, Ptomacanthus demonstrates an osteichthyan-like pharynx letting authors to suggest the ancestral gnathostome branchial skeleton condition. Moreover, the new data address phylogenetic relationship of chondrichthyan stem group nesting Gladbachus as the sister group of chondrichthyan crown group and all the "acanthodian grade"

as a paraphyletic basally to (Gladbachus + chondrichthyan). I agree with this arrangement, more "natural" than some previous proposals

Overall, I would recommend this paper for publication in Nature Comms. It would be of high-interest to a broad audience and materially contributes to a number of significant debates about early gnathostome relationships.

SPECIFIC COMMENTS

Line 36 Abstract.

"Ptomacanthus, a 415 million year old chondrichthyan"

I recommend refer to Ptomacanthus as STEM chondrichthyan (here an throughout the entire paper). Strictly acanthodian grade stem-chondrichthyans lacks several of the sinapomorphie defining Chondrichthyans. (some of them listed on the paper (pag. 7 line 216). Lacking a "formal" nomenclatural term for ("acanthodians" + Chondrichthyan) better refer the first as Stem-Chondrichthyans.

Line 41 Abstract

"challenging the idea that the ancestral gnathostome pharynx conformed to a strict morphological archetype".

This sentence is vague. Better finished abstract with a short sentence synthetizing the ancestral gnathostome branchial skeleton reconstructed by the authors (e.g. lines 295-314 and other parts of the mns.).

Line 89

Recent investigations using computed tomography scanning have revealed a spate of new data on the gill skeletons of early gnathostomes 5,6,15 However, chondrichthyan examples display already elasmobranch-like pharyngeal skeletons".

At this point of the paper (Introduction) Reference 5 claimed the presence of "osteichthyan-like branchial arches in a Palaeozoic shark (Ozarcus)". This is view is followed in ref 6.

If authors contradict this interpretation in the mns (as it is done) this must be argued in the discussion but not assume in the introduction.

Authors need to be more specific why they interpreted Ozarcus pharynx as elasmobranch-like contra previous authors considering it as osteichthyan-like.

In the same sense symmoriiiforms are recovered as stem-chondrichthyan by Pradel et al. while they are nested as stem-holocephalans by the authors analysis.

Line 173 Phylogenetic analysis

Authors sustain the use of the Adams consensus tree to find some resolution of the chondrichthyan stem-group (not found in strict consensus tree). However in Line 393 it is sustained that a majority rule consensus tree with a relative burnin of 25% is calculate. (Write burn-in)

Supplementary info and supplementary figure s4 also sustained use of to the majority rule consensus tree.

Please be clear which consensus tree you calculate. Anyway take into account that Usually n=50 and referred to as majority rule consensus.

If Aam consensus tree is calculate noted that Adams consensus trees can include groups that don't occur in any input tree and they could weakly supportedd

Line 215

"the chondrichthyan crown-group to the exclusion of acanthodian-grade taxa, including extensively calcified cartilage, the absence of any kind of large dermal cranial ossification, and the absence of paired fin spines".

Paired pectoral fin-spines are known in Dolious (Miler et al 3003)

FIGURE 3 caption

"... O indicates the appearance or loss of a single operculum...!"

There is not O in the figure

"... Colour scheme: gray, neurocranium; lightyellow, hyomandibular....."

Please check the colors. I can distinguish dark or orange but brown an green

It is Ozarcus the best representative for chondrichthyan crown-group??

FIGURE 4 caption

"..Colour scheme: as in figure 6.."

There is not Figure 6

Many thanks to the reviewers for their helpful comments. We have made changes to the manuscript, and have given replies to comments in the text below. In addition to the changes in response to the referees, we have added a co-author as we thought the work contributed by a summer research student warranted co-authorship on this work.

Reviewer #1 (Remarks to the Author):

This manuscript provides the first description of the visceral (branchial) skeleton of the 'acanthodian'-grade stem chondrichthyan *Ptomacanthus* from the Lower Devonian of UK. These data on 'acanthodians' were eagerly awaited in the framework of current debates about the origin of chondrichthyans and the plesiomorphic condition of the branchial skeleton in jawed vertebrates. The paper is excellently executed, and the segmentations of the CTscan images is a real technical feat, especially considering the low degree of mineralization of the perichondral lining of the elements of the branchial skeleton in these taxa. The authors provide a very interesting interpretation of the timing of the process of posterior extension of the branchial skeleton at the node "between" *Ptomacanthus* and *Gladbachus*. The conclusion is that the plesiomorphic pattern for the crown gnathostome pharynx may have been a mix of osteichthyan-like and chondrichthyan-like patterns. I would agree entirely.

Minor corrections:

p.8, l. 254: "While the condition in could be equally parsimonious..." in what? A word seems to be missing.

This has been addressed by rewording this sentence.

Caption to figure 3: "anterior extent, O indicates the appearance..." I don't see this "O" in the figure

This has been corrected in the figure

Caption to fig. 4: "(h) Debeerius"... isn't it rather (l), as (h) already appears in the line above for *Triodus*?

This is correct, and has been fixed

Reviewer #2 (Remarks to the Author):

The gill skeleton is a key vertebrate anatomical system which provides a wealth of phylogenetic data to understand the evolution of early gnathostomes, as well as in debates on the origin of jaws or paired appendages. However, due to weak mineralization the fossil data on the gill skeleton of early gnathostomes are comparatively few, limiting our understanding of their character polarity or leaving considerable uncertainty on the ancestral complement of elements in the gnathostome gill skeleton. The description of the gill skeleton of *Ptomacanthus*, the earliest three-dimensionally articulated branchial skeleton known in a gnathostome, will help to constrain some major changes in pharyngeal architecture close to the split between osteichthyans and chondrichthyans or close to the chondrichthyan crown node. The general morphology of gill skeleton in *Ptomacanthus* is well described and very informative. As such, I recommend it to be accepted with minor revisions.

Meanwhile, I wonder that the authors should be careful of the discussions on the 'archetype', which is out-dated, and more like a jackstraw. With the updated phylogenetic scenario of early jawed vertebrates, it is normal that the previous understanding on the ancestral character combination is under change, along with the fast-emerging evidences, either from the new fossils or by means of new technique as in this study.

We agree with the reviewer that ideas of an "archetype" are outdated, and the majority of modern palaeontologists (including us) would expect to see combinations of plesiomorphic and derived characters in an animal like *Ptomacanthus*. However, as we highlight in the main text, outside palaeontology the idea of sharks as "primitive" persists, and we would hope that the broad reach of this paper in *Nature Communications* will be of interest to people in disciplines such as developmental biology as well as the general public who may not expect this. We have also removed the word archetype in the introduction (second paragraph) to clarify this.

Regarding data accessibility, I recommend the authors to upload the stacks and all scan parameters to a public platform for download, if possible. This will help readers examine the digital restorations, which form the cornerstone of this manuscript.

We agree with the reviewer and will include a figshare address with the stacks, scan parameters, and models with the final submission.

I.31-32: "However little is known about the gill skeletons of early chondrichthyans....."

Note that the gill skeleton is well described in one of acanthodians (acanthodian-grade stem chondrichthyans), Acanthodes, and some early conventionally-defined chondrichthyans, such as *Gladbachus* and *Ozarcus*.

We have addressed this in the abstract by changing the wording.

I.38: "a combination of chondrichthyan-like and plesiomorphic conditions" as "a combination of chondrichthyan-like and plesiomorphic gnathostome conditions".

This has been added.

I.116: “the ceratohyal of the fifth arch” Do you mean the ceratobranchial of the fifth arch?
Yes, we did. Changed accordingly

I.183: Gladbachus in italics.
Changed

I.254: “While the condition in could be”; in what?
Addressed: see reply to Reviewer #1’s comments

I.246-259: The arguments in this paragraph are important but not so solid. As acknowledged by the authors, the posterior extension in Acanthodes and Gladbachus + crown chondrichthyans can be equally resolved as one innovation at the node TG chondrichthyans and one reversal in Ptomacanthus. The argument “a seemingly compact condition in the acanthodian Diplacanthus” is weak, considering many other acanthodians with the same phylogenetic grade as Acanthodes have the similar elongated opercular area (as indicated by the long branchiostegals, such as in Euthacanthus and Mesacanthus) to Acanthodes. The resolution of this uncertainty of character polarity needs additional fossil data close the crown-gnathostome node. This is beyond the scope of this study.

We consider that while it is true that these taxa (eg. *Mesacanthus* and *Euthacanthus*) may have extended pharynxes, the taphonomy of the fossils (ie. the flattened condition of them) makes it difficult to judge the exact relative positions of dermal structures of the head (the only preserved part) and internal structures like the pharynx. Part of what makes *Ptomacanthus* such an important source of information is that the dimensions/position of the pharynx can be understood from reference to both internal structures and dermal structures of the head skeleton. Nonetheless we have addressed this with changes to the text in this section, acknowledging these taxa and that the taphonomy of these fossils makes their interpretation difficult.

I.260-278: The presence or absence of a bony operculum is one of main topics in this manuscript. It is very helpful to readers to add some photos or illustrations of the bony operculum in Ptomacanthus and Acanthodes, at least in supplementary information.

We agree this would be useful, and have labelled the position of the bony operculum on the mould of *Ptomacanthus* in Figure 1 and Supplementary figure 3.

Figure 3. No ‘O’ in the illustration.
Fixed: see reply to Reviewer #1

Figure 4: “as in figure 6”? Do you mean “as in Figure 2”?
Yes we do: fixed

I.311-314: You need to rephrase this sentence.
This sentence has been rephrased

In Figures S1-S3, the scale bars need to be added, as in Figures 1 and 2.

This has been done

Min Zhu

Reviewer #3 (Remarks to the Author):

GENERAL COMMENT

Dear Authors,

I find this MS interesting and the studies well performed.

The manuscript details a study of the gill skeletons preserved in a fossil (415 m.y. old) STEM chondrichthyan (*Ptomacanthus*). This specimen was studied in the past by several authors and demonstrated as a key fossil to understand polarity of characters and interrelationships of mayor groups of early gnathostomates. In the present mns. Dearden & Brazeau use computed tomography scanning to study new anatomical structures in the specimen (i.e. the branchial apparatus). The new data contribute to change the erroneous view of extant chondrichthyans as indirect means for plesiomorphic gnathostomate pharynx conditions. Contrarily, *Ptomacanthus* demonstrates an osteichthyan-like pharynx letting authors to suggest the ancestral gnathostome branchial skeleton condition. Moreover, the new data address phylogenetic relationship of chondrichthyan stem group nesting Gladbachus as the sister group of chondrichthyan crown group and all the “acanthodian grade” as a paraphyletic basally to

(Gladbachus + chondrichthyan). I agree with this arrangement, more “natural” than some previous proposal

Overall, I would recommend this paper for publication in Nature Comms. It would be of high-interest to a broad audience and materially contributes to a number of significant debates about early gnathostome relationships.

SPECIFIC COMMENTS

Line 36 Abstract.

“*Ptomacanthus*, a 415 million year old chondrichthyan”

I recommend refer to *Ptomacanthus* as STEM chondrichthyan (here an throughout the entire paper). Strictly acanthodian grade stem-chondrichthyans lacks several of the sinapomorfie defining Chondrichthyans. (some of them listed on the paper (pag. 7 line 216). Lacking a “formal” nomenclatural term for (“acanthodians” + Chondrichthyan) better refer the first as Stem-Chondrichthyans.

We have made the recommended change in the abstract, and have checked through the paper for similar uses. We also prefer a Total Group Concept based definition of chondrichthyans (ie. stem- + crown- = total-group) , rather than the reviewers character-based definition.

Line 41 Abstract

“challenging the idea that the ancestral gnathostome pharynx conformed to a strict morphological archetype”. This sentence is vague. Better finished abstract with a short sentence synthetizing the ancestral gnathostome branchial skeleton reconstructed by the authors (e.g. lines 295-314 an other parts of the mns.).

We have rewritten the final part of the abstract according to the reviewer's suggestion

Line 89

Recent investigations using computed tomography scanning have revealed a spate of new data on the gill skeletons of early gnathostomes 5,6,15 However, chondrichthyan examples display already elasmobranch-like pharyngeal skeletons”.

At this point of the paper (Introduction) Reference 5 claimed the presence of “osteichthyan-like branchial arches in a Palaeozoic shark (*Ozarcus*)”. This is view is followed in ref 6.

If authors contradict this interpretation in the mns (as it is done) this must be argued in the discussion but not assume in the introduction.

Authors need to be more specific why they interpreted *Ozarcus* pharynx as elasmobranch-like contra previous authors considering it as osteichthyan-like.

In the same sense symmoriiforms are recovered as stem-chondrichthyan by Pradel et al. while they are nested as stem-holocephalans by the authors analysis.

Where we refer to *Ozarcus* as “elasmobranch-like” in this section, we really meant insofar as it has an extended pharynx, rather than in terms of patterning (which is what Pradel *et al.* argue is osteichthyan-like). However, this was a bit ambiguous, and we have emended the text of the introduction to clarify the point.

Line 173 Phylogenetic analysis

Authors sustain the use of the Adams consensus tree to find some resolution of the chondrichthyan stem-group (not found in strict consensus tree). However in Line 393 it is sustained that a majority rule consensus tree with a relative burnin of 25% is calculate. (Write burn-in)

Supplementary info and supplementary figure s4 also sustained use of to the majority rule consensus tree.

Please be clear which consensus tree you calculate. Anyway take into account that Usually n=50 and referred to as majority rule consensus.

We ran two separate phylogenetic analyses: a parsimony analysis and a Bayesian analysis. The Adams consensus tree is only used for the parsimony results, while the majority rule consensus is only used for the Bayesian results. Part of this confusion arose from a typo in the caption for Figure S4, which has been fixed, and the caption for Figure S5 has also been edited for clarification. We have also corrected the spelling of burn-in.

If Aam consensus tree is calculate noted that Adams consensus trees can include groups that don't occur in any input tree and they could weakly supportedd

The clusters in an Adams tree do not necessarily reflect phylogenetic relationships, but the resolved nodes represent strictly conserved branching patterns within the fundamental tree set. For explorations of key taxa (*Ptomacanthus*, *Gladbachus*, and crown-chondrichthyans), this suffices to show relative branching arrangements that exist in all trees but do not show in the strict consensus, as it is insensitive to these patterns (Wilkinson 1994 ).

Line 215

“the chondrichthyan crown-group to the exclusion of acanthodian-grade taxa, including extensively calcified cartilage, the absence of any kind of large dermal cranial ossification, and the absence of paired fin spines”.

Paired pectoral fin-spines are known in *Doliodus* (Miler et al 3003)

This is true, and *Doliodus* is coded as such in our matrix. However, as subsequent work on *Doliodus* demonstrates (eg. Maisey et al. 2017, Long et al. 2015), the animal actually shares many characteristics with “acanthodian-grade” taxa, and in the above sentence is really intended to be included within the “acanthodian grade”. As noted above, we use a Total-group based, rather than synapomorphy-based, definition of chondrichthyans.

FIGURE 3 caption

“... O indicates the appearance or loss of a single operculum...!”

There is not O in the figure

Fixed: see reply to Reviewer #1

“.... Colour scheme: gray, neurocranium; lightyellow, hyomandibular.....”

Please check the colors. I can distinguish dark or orange but brown an green

We have modified the colour scheme and descriptions to make this clearer

It is *Ozarcus* the best representative for chondrichthyan crown-group??

We agree that *Ozarcus* is not the surest member of the chondrichthyan crown-group, given disagreements on the placement of symmoriiiformes. However, it is nonetheless an extremely important reference point for early chondrichthyan gill skeletons, as the best-preserved Palaeozoic “shark” in which both the braincase and branchial skeleton are well-known and described in detail. Given figures 3 and 4 (and our interpretation of gnathostome gill evolution) are based on the topology we recover in our phylogenetic analysis, we feel justified in including *Ozarcus* where we recover it in this topology. Furthermore in figure 4, where it is the sole representative of crown-chondrichthyans, it show identical conditions to living elasmobranchs such that if *Ozarcus* did prove to be a stem-chondrichthyan conclusions drawn from the figure would be the same.

FIGURE 4 caption

“..Colour scheme: as in figure 6..”

There is not Figure 6

Fixed: see reply to Reviewer #2

Reviewers' Comments:

Reviewer #3:

Remarks to the Author:

I consider that in general the points raised in the previous round of review have been satisfactorily addressed.